# Bioactive Compounds and Antioxidant Capacity of Several Blackberry (*Rubus* spp.) Fruits Cultivars Grown in Romania

Adriana Ramona Memete [1,†] , Ioan Sărac [2] , Alin Cristian Teusdea [1,†] , Ruben Budău [1,*] , Mariana Bei [1] and Simona Ioana Vicas [1,*]

1    Faculty of Environmental Protection, University of Oradea, 26 Gen. Magheru Street, 410048 Oradea, Romania; memeteadriana25@gmail.com (A.R.M.); mbei@uoradea.ro (M.B.)
2    Faculty of Engineering and Applied Technologies, Banat's University of Agricultural Science and Veterinary Medicine "King Michael I of Romania" from Timisoara, Calea Aradului 119, 300645 Timișoara, Romania; ioansarac@usab-tm.ro
*    Correspondence: rbudau@uoradea.ro (R.B.); svicas@uoradea.ro (S.I.V.)
†    These authors contributed equally to this work.

**Abstract:** Blackberry fruit (*Rubus* spp.) has a powerful antioxidant capacity due to the high levels of anthocyanins and other phenols it contains. In this work, we investigated the physico-chemical characteristics, the bioactive compounds (total phenols, flavonoids, and monomeric anthocyanins), and the antioxidant capacity of seven blackberry cultivars belonging to the *Rubus fruticosus* L. and *Rubus laciniatus* L. genera growing in the NW region of Romania. In addition, the wild blackberry cultivar from the same area was also evaluated. Anthocyanins from the blackberry fruit were extracted using SPE (Solid Phase Extraction), and the anthocyanin profile was identified and quantified using HPLC-PDA analysis. In terms of polyphenol content and antioxidant capacity, two of the cultivars examined stood out. The majority of anthocyanin found in blackberries was cyanidin-3-glucoside, with the highest amount recorded in the 'Thorn Free' cultivar at $329.26 \pm 9.36$ mg/g dw. Comparatively, 'Loch Ness' and 'Thorn Free' fruits exhibited total phenol contents of $1830.98 \pm 13.55$ and $1687.14 \pm 62.41$ mg GAE/100 g dw, respectively. Antioxidant capacities varied significantly among the eight blackberry cultivars, with cultivars 'Loch Ness' and 'Thorn Free' achieving high values in comparison to the others. The findings of the multivariate analysis also supported the experimental results. Knowing the phytochemical composition and antioxidant potential of different blueberry cultivars, one can use them as fresh, functional foods or for commercial purposes to produce products derived from fruits with a high concentration of bioactive components.

**Keywords:** *Rubus fruticosus* L.; *Rubus laciniatus* L.; anthocyanins; total phenols; flavonoids; antioxidant capacity; multivariate analysis





## 1. Introduction

Blackberry is a shrub belonging to the genus Rubus of the family *Rosaceae*. The family *Rosaceae* is the 19th largest family of plants. Rubus, with almost 700 species, is the largest genus in this family. Rubus comprises 12 subgenera, with few domesticated species. It is a perennial plant believed to have originated in Armenia, with rapid growth distributed and cultivated mainly in Europe, Asia, and North America, but its worldwide popularity is constantly growing. It grows spontaneously in the northern parts of Pakistan, where it is known by various local names: Karwara, Ach, Akhara, and Baganrra. Although the fruit is widely accepted in Pakistan, it is not grown on a commercial scale. There were 7692 ha of blackberries commercially grown in Europe in 2005, with Serbia leading as a producer, followed by Hungary, the United Kingdom, Romania, Poland, Germany, and Croatia [1]. In recent years, Europe has become one of the world leaders in the production of blackberries (*Rubus fruticosus*); the largest regions with blackberry production are Serbia and Hungary [2–6].

Because blackberries naturally grow in relatively vast regions in Romania and the country's primary source of blackberries comes from this flora, blackberries are only grown there on small plots of land. However, cultivating blackberries has attracted particular attention in Romania in recent years. Thus, blackberries harvested from spontaneous flora are of great interest among consumers, and the fruits are consumed both fresh and in the form of juices, jams, compote, syrup, or in the preparation of desserts.

In 2005, the area of wild blackberries in Romania was approximately 2400 ha, and the cultivated organic blackberry area was only 10 ha. Of the cultivated varieties, Loch Ness is the main variety cultivated in Romania [1]. The blackberry of the spontaneous flora produces edible fruits and is spread all over the world, but is mainly concentrated in the northern hemisphere. Blackberries were among the first fruits of the wild flora used for medicinal purposes. Blackberry juice has been used in Europe since the 16th century to treat various oral and eye infections, and the cultivation of blackberry species began in the 17th century [2,7]. The global area under blackberry cultivation expanded by nearly 45% between 1997 and 2007 [5].

Favorable weather and soil conditions, as well as the experience of growers, promote the development of crops. Farmers have developed a variety of *R. fruticosus* varieties using traditional breeding procedures, which differ in firmness, shape, size, aroma, color, weight, yield, ripening season, nutritional content, and pest resistance. The most popular varieties include Jumbo, Chester, Bartin, Ness, Bursa 1, Bursa 2, Bursa 3, Arapaho, Navaho, Thornfree, Chester Thornless, Dirksen Thornless, Cacanska Bestrna, Loch Ness, Cherokee, and Black Satin [3,6].

Fruits from the *Rubus* genus are among those rich in bioactive compounds. Their nutritional profile shows that they contain dietary fiber, vitamins, minerals, and carbohydrates [2,8,9]. In general, fruits have a high level of dietary fiber (cellulose, hemicellulose, pectin) but are low in calories, very low in fats, and high in carbohydrates such as glucose and fructose. Organic acids are present in low concentrations and are responsible for the flavor of the fruit. Certain minerals, some vitamins and the presence of phenolic compounds give Rubus fruits the status of functional foods [4,10,11]. Blackberries have a variety of health benefits due to their phytochemical composition, which helps prevent and treat metabolic disorders and chronic diseases such as cancer, diabetes, hypertension, cardiovascular disease, gastrointestinal diseases, atherosclerosis, aging, Parkinson's disease, and Alzheimer's disease [6,12,13].

Anthocyanins, in particular, provide a considerable contribution to blackberries' antioxidant potential and are responsible for a number of benefits for both human and animal health, according to recent literature data on their phytochemical composition [8,9,11,14,15]. Along with flavonoids—such as various glycosylated forms of quercetin, kaempferol, luteolin -3-O-glucuronide, and apigenin -3-glucuronide—cyanidin-3-O-glucoside is the major anthocyanin that predominates among the compounds found in Rubus fruits. Various other anthocyanins are also detected in blackberry fruits, such as cyanidin-3-O-arabinoside, cyanidin-3-O-xyloside, cyanidin-3-O-malonylglucoside, cyanidin-3-O-dioxalylglucoside, cyanidin-3-O-diglucoside, cyanidin-3-glucosylrutinoside, cyanidin-3-O-rutinoside, cyanidin-3-(3′-malonyl)glucoside, and cyanidin-3-(6′-malonyl)glucoside. In addition, anthocyanins, a water-soluble pigment, have been found in fruit cell vacuoles [13,15]. Anthocyanins are known to be powerful antioxidants that have the ability to fight oxidative stress and eliminate free radicals from the body, and supplementing the diet with natural antioxidants obtained from fruits could be more effective than consuming an individual antioxidant obtained from other sources [16–18]. The daily intake of anthocyanins in humans in the United States was estimated by Kuhnau, 1976, to be between 180 and 215 mg/day [19]. Blackberries also contain appreciable amounts of flavonols that appear in the glycosylated form and are found exclusively in the fleshy part of the fruit [20].

The objective of this study was to characterize the physical and chemical properties, and total bioactive compound content of blackberry fruits cultivated on a farm in northwest Romania. In addition, the blackberry antioxidant capacity was also investigated. Beyond

these blackberry cultivars, the wild blackberry variety was investigated. Multivariate statistical analysis was used to analyze the results and identify the most valuable blackberry cultivar in terms of its bioactive compounds level and antioxidant capacity. The novelty of this study lies in the fact that blackberry fruit cultivars were characterized in the literature for the first time in terms of their bioactive compounds and antioxidant capacity. Our research focused on the *Rubus fruticosus* L. ('Cester,' 'Triple Crown,' 'Navaho,' 'Loch Ness,' 'Thorn Free,' and 'Ouachita,' including the wild variant) and *Rubus laciniatus* L. ('Thornless Evergreen') genus.

## 2. Materials and Methods

### 2.1. Chemicals

The HPLC reference standards of cyanidin chloride (purity $\geq$ 95% HPLC) were purchased from Sigma Aldrich (St. Louis, MO, USA). The water for the HPLC analysis was purified using a Milli-Q system (Merck Millipore, Burlington, MA, USA). For spectrophotometric methods, quercetin, 6-hydroxy-2,5,7,8-tetramethylchroman-2-carboxylic acid, Folin and Ciocalteu's reagent were purchased from Sigma Aldrich (St. Louis, MO, USA), and 2,4,6-Tris(2-pyridyl)-s-triazine and aluminum chloride were purchased from Fluka (Charlotte, NA, USA). Neocuproine Hemihydrate $\geq$ 99%, p.a. was purchased from Roth. Ethanol, methanol, copper chloride, and sodium hydroxide were of analytical grade.

### 2.2. Plant Materials

Seven different cultivars of blackberries were harvested in 2021 from the Bărzani farm in Arad County, located in northwest Romania (coordinates: 46°29'7'' N 22°7'57'' E), while wild blackberry fruit was harvested from the region's (northwest Romania) native vegetation. Blackberries naturally grow in Romania in plain areas, on steep slopes, and in forested areas. Blackberries from the spontaneous flora (WILD) are thorny shrubs and have very few fruits in comparison to those from cultivated plants [21].

The cultivated blackberries used in this study are used for human consumption and belong to both the genus *Rubus fruticosus* L. ('Chester', 'Triple Crown', 'Navaho', 'Loch Ness', 'Thornfree' and 'Ouachita') and *Rubus laciniatus* ('Thornless Evergreen'), and are described in Table 1.

**Table 1.** Horticultural characterization and the appearance of blackberry cultivars.

| Cultivars | Aspect of Fruits | General Aspects |
|---|---|---|
| CHEST |  | Is a vigorous shrub, semi-erect and fast-growing:<br>❖ Height of the plants: 3–4 m;<br>❖ Ripening: August–September;<br>❖ Frost resistance: of all the thornless varieties, it is the most resistant;<br>❖ The fruits: produced on shoots of the second year; they are large and sweet, with an aromatic taste and a uniform ripening;<br>❖ Productivity: approximately 5–6 kg/plant or 15–18 t/ha. |
| T_C |  | Is a semi-erect blackberry variety and self-fertile:<br>❖ Height of the plants: max. 3 m;<br>❖ Ripening: 30 days between the months of July and August;<br>❖ Frost resistance: −20 °C (The frost resistance is average, compared to 'Cester');<br>❖ The fruits: large, firm, and with extremely high productivity;<br>❖ Productivity: approximately 15 t/ha. |

**Table 1.** *Cont.*

| Cultivars | Aspect of Fruits | General Aspects |
| --- | --- | --- |
| NAV |  | Is an erect shrub, and prefers sunny areas and less wet and cold soils:<br>❖ Height of the plants: approximately 2 m;<br>❖ Ripening: 30 days, in June;<br>❖ Frost resistance: −9 °C;<br>❖ The fruits: produced on second-year shoots, large, firm, and juicy;<br>❖ Productivity: highly productive. |
| L_NESS |  | Is a vigorous shrub with semi-erect shoots, prefers very sunny areas and is particularly suitable for smaller gardens as it is compact and does not produce suckers:<br>❖ Height of the plants: 3–4 m;<br>❖ Ripening: starts from the end of July and ends in September;<br>❖ Frost resistance: −25 °C;<br>❖ The fruits: medium-sized, firm, cone-shaped, and ripen evenly;<br>❖ Productivity: 3.6 kg of fruit/plant. |
| THRNFR |  | Is a very vigorous shrub, and during the vegetation period, they produce side shoots; the inflorescences are long, with a large number of flowers:<br>❖ Height of the plants: 4–8 m;<br>❖ Ripening: is prolonged and staggered, beginning in mid-August and extending to early September;<br>❖ Frost resistance: has good resistance to frost;<br>❖ The fruits: very large fruits, pleasant and slightly aromatic taste, and appear only in the 2–3 years after planting;<br>❖ Productivity: approximately 20 t/ha. |
| OCHIT |  | Is a very vigorous, fast-growing, upright, and erect shrub, and is a sun lover and self-pollinating:<br>❖ Ripening: begins at the end of June and extends over a period of 5 weeks;<br>❖ Frost resistance: moderate; it prefers areas with a milder climate;<br>❖ The fruits: produced on shoots of the second year, are medium to large in size, and extremely sweet and juicy, rivaling the NAV variety;<br>❖ Productivity: approximately 3 kg/plant. |
| EVRG |  | Is a semi-vigorous and productive variety, vulnerable to various diseases, very resistant to drought conditions, and its leaves remain green even in winter:<br>❖ Height of the plants: the stems are creeping and long—3–6 m;<br>❖ Ripening: from July to September;<br>❖ Frost resistance: is sensitive to frost but tolerates cold;<br>❖ The fruits: have a juicy pulp with a sweet taste in adulthood, firm, and resistant to transportation;<br>❖ Productivity: 12–14 t/ha. |

CHES—'Chester;' T_C—'Triple Crown;' NAV—'Navaho;' L_NESS—'Loch Ness;' THRNFR—'Thornfree;' OCHIT—'Ouachita;' EVRG—'Thornless Evergreen.'

All the fruits were collected at optimum full ripeness. Berries were handpicked and transported (ca. 2 h) in cooled containers to the Food Engineering Laboratory of the University of Oradea. The blackberry fruits were divided into two groups, one group being used for physico-chemical analyses (fruits size and weight, total soluble solids, pH, and titratable acidity), and another group was dried at 40 °C in the oven with ventilation (Nitech Pol Eko oven, model CLN 53, Wodzisław, Poland) for a long time (a few days, depending on the variety) until a constant weight was reached. The dried blackberries were ground to a fine powder, packed in plastic bags and stored in the dark before extraction for the following analysis described below. According to the literature, dehydrating fruit at a temperature of 40 °C has been shown to prevent the deterioration of the compounds that are present in fruits, particularly the anthocyanin content [22].

### 2.3. Determination of Fruit Weight, Dimensions, Moisture and Firmness of Blackberries

Thirty fruits from each variety were used for the determination of weight and size. Fruit weight was measured with an analytical balance with a sensitivity of 0.0001 g, and length and width were measured with a 150 mm digital caliper with a sensitivity of 0.01 mm (RS PRO, China).

By monitoring the weight loss of oven-dried blackberries at 40 °C (until they reached a constant weight), the moisture content was determined using Equation (1) [23].

$$\%\text{Moisture} = \frac{Wi - Wf}{Wi} \times 100 \tag{1}$$

where *Wi* is the weight of fresh fruits; *Wf* is the weight of dried fruits.

Firmness was measured on two opposite sides of 30 fruits per treatment using a portable penetrometer (Fruit penetrometers tester, FT 327) with a 3 mm diameter probe, and the average value was expressed in Newton (N). The firmness of the blackberries was measured immediately after harvest.

### 2.4. Determination of Total Soluble Solids Content (TSS), pH and Titratable Acidity of Blackberries

TSS was determined from homogenized fruits at 20 °C using a digital Abbe refractometer. After diluting 10 g of the homogenized material with 100 mL of deionized water, the pH was measured using a digital pH meter. Titratable acidity was determined by the titrimetric method using 0.1 N NaOH solution, and the results were expressed as g citric acid/100 g fw.

### 2.5. The Bioactive Compounds Determination

#### 2.5.1. Preparation of Blackberry Extracts

1 g of the powder obtained from dehydrated blackberries in the oven (Nitech Pol Eko oven, model CLN 53, Poland) at 40 °C was extracted with acidified methanol (trifluoroacetic acid (TFA), 0.1%) in a ratio of 1:10 (*w/v*), using the Heidolph homogenizer, Silent Crusher M at a speed of 12,000 rpm, for one minute. The homogenized samples were then centrifuged (NÜVE NF 200 BENCH TOP CENTRIFUGE, Ankara, Turkey) at 5000 rpm for 20 min. The pellet was re-extracted until the solvent became colorless, then supernatants were combined, and the methanol was removed by vacuum rotary evaporator (Heidolph Rotary Evaporator, Laborota 4000 rotavapor, Schwabach, Germany) at 40 °C, according to Bunea, et al., (2011), and the dry residue was dissolved in acidified water (0.1% TFA) [24,25]. Finally, bioactive compounds-rich extract was produced, and total polyphenols, flavonoids, monomeric anthocyanins content and antioxidant capacity were measured.

#### 2.5.2. Determination of Total Phenols (TPh), Flavonoids (TFlav) and Total Monomeric Anthocyanins (MAP) Content

The total phenols content in the blackberry fruits was determined using the Folin–Ciocalteu method with some modification [26,27]. Briefly, the aqueous blackberry extract (100 μL) was incubated with 1700 μL of distilled water, 200 μL of Folin–Ciocalteu reagent

(freshly prepared, dilution 1:10, $v/v$) and 1000 µL of 7.5% $Na_2CO_3$ solution for 2 h in the dark, at room temperature. The absorbance was measured at 765 nm (Shimatzu miniUV-Vis spectrophotometer, Kyoto, Japan), and the results are expressed as milligrams of gallic acid equivalents (GAE) per 100 g dry weight (dw) using gallic acid as a standard.

The total flavonoid content was determined by the colorimetric method of aluminum chloride. Aqueous blackberry extract (1 mL), 4 mL distilled water, and 0.3 mL 5% $NaNO_2$ were combined in a 10 mL volumetric flask and allowed to react for 5 min. After that, 0.3 mL of 10% $AlCl_3$ was added, and the mixture was homogenized before a 6 min break. A final volume of 10 mL was obtained by adding 2 mL of 1 M NaOH and distilled water, and after 15 min of pause, a spectrophotometer was used to measure the absorbance at a wavelength of 510 nm. The results were expressed in mg QE (quercetin equivalents)/g dw [28].

Total monomeric anthocyanin pigments content in blackberry fruits was determined by the pH-differential method [29,30], and the results were expressed as cyanidin-3-glucoside equivalent (C3glu)/100 g dw.

### 2.6. Solid Phase Extraction and HPLC-PDA Analysis of Anthocyanins

As schematically illustrated in Figure 1, the solid phase extraction (SPE) of blackberries anthocyanins was carried out. The blackberry extract was passed through SPE C18 column (200 mg/3 mL) (Finisterre by Teknokroma, Barcelona, Spain)The remaining compounds in the sorbent were washed with water acidified with 0.01% HCl, and then the anthocyanins were eluted with 10 mL of acidified methanol (0.01% HCl). The rotavapor was used to evaporate the methanol from the anthocyanin extract. The aqueous extract obtained was dissolved with 0.01% HCl acidified water to give the anthocyanin-rich extract (R.A.E.) [31].

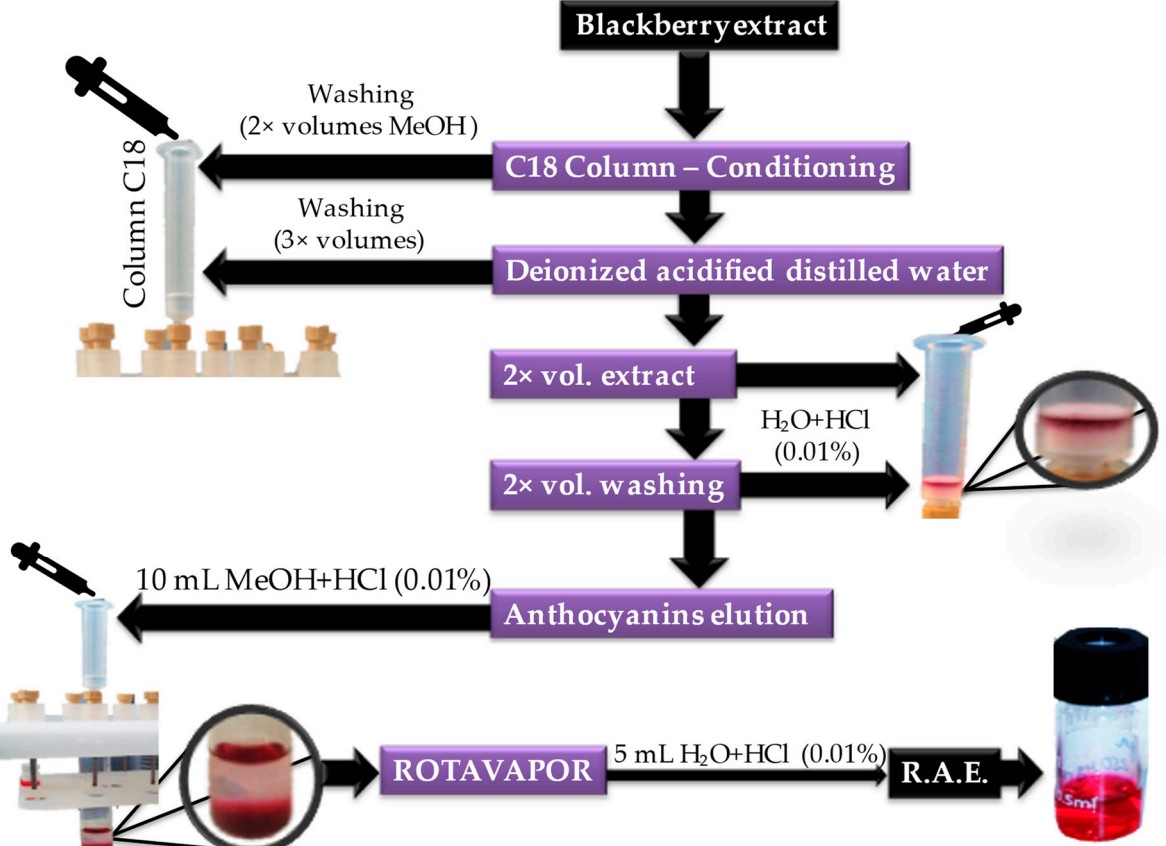

**Figure 1.** Extraction and purification of anthocyanins from blackberry cultivars using SPE. R.A.E. (rich anthocyanin extract).

Chromatographic analysis of anthocyanins from blackberry fruits was carried out using an HPLC system (Shimadzu Corporation, Scientific Instruments, Kyoto, Japan) equipped with a CBM-20A controller, an LC-20AD pump, a DGU-20A degasser, a SIL-20AC, a CTO-20AC column oven, and a photodiode array detector (SPD-M20A, PDA). The chromatographic column used was Luna (C18) (Phenomenex, Torrance, CA, USA) 100 Å, (250 mm$\times$ 4.6 mm, 5 µm), at 30 °C, using a flow rate of 0.5 mL/min and a sample injection of 20 µL. The obtained data were processed by the Labsolution program version 5.10.153 (Shimadzu). Mobile phases were made up of eluent A (4.5% formic acid solution) and eluent B (acetonitrile) using the following gradient schedule: 0–9 min 10% B; 10–17 min linear gradient up to 12% B; 18–30 min linear gradient up to 25% B; 31–50 min linear gradient up to 90% B; and 51–55 min linear gradient up to 10% B [31].

Anthocyanins were detected by monitoring the absorbance at 520 nm. Both the retention time and the UV spectrum of each obtained peak were compared with that of the standards and with the Polyphenol Explorer database (http://phenol-explorer.eu/compounds/classification, accessed on 16 September 2021).

A calibration curve was created using cyanidin chloride (purity $\geq$ 95% HPLC, Sigma) as the standard at concentrations ranging from 15 to 500 µg/mL in order to quantify anthocyanin compounds. Results are expressed as mg cyanidin equivalent/g dw. The regression equation was y = 3681.7x + 1654.3 ($R^2$ = 0.9967).

### 2.7. Determination of Antioxidant Capacity of Blackberries

Antioxidant capacity of blackberry samples was determined using four spectrophotometric assays, DPPH (2.2, diphenyl-picryl-hydrazyl), FRAP (Ferric Reducing Antioxidant Power) and CUPRAC (Cupric Reducing Antioxidant Capacity).

Radical scavenging capacity of blackberry extract using the stable DPPH radical was determined according to the method of Brand Williams, et al., 1995, with minor modifications [27,32]. A volume of 100 µL of blackberry extract was mixed with 2800 µL of 80 µM DPPH solution and stored at room temperature in the dark. After 30 min, the absorbance of the samples was measured using a Shimadzu mini UV-VIS spectrophotometer at a wavelength of 517 nm. The results were expressed as mmol Trolox equivalent (TE)/100 g dw. The regression equation was y = 970.07x + 0.6222 ($R^2$ = 0.998).

The FRAP assay tested the antioxidant power of the blackberry samples based on the extract's ability to reduce $Fe^{3+}$ from tripyridyltriazine $Fe(TPTZ)^{3+}$ complex to the blue-colored complex-$Fe(TPTZ)^{3+}$ in an acidic medium [33,34]. Blackberry extract (100 µL) was allowed to react with 500 µL FRAP working solution and 2 mL distilled water, for 1 h, in the dark. The results were expressed in mmolTrolox equivalent (TE)/100 g dw. The regression equation was y = 13,588x + 0.046 ($R^2$ = 0.9977).

The CUPRAC method consists of the combination of 1 mL copper (II) chloride solution ($1 \times 10^{-2}$ M), 1 mL neocuproine (2.9-dimethyl-1, 10-phenanthroline) alcoholic solution ($7.5 \times 10^{-3}$ M), 1 mL ammonium acetate aqueous buffer (pH 7), and 100 µL blackberry extract, followed by water such that to make the final volume 4.1 mL. The samples' absorbance was measured at 450 nm following a 30-min pause in the dark. The results were expressed in mmol TE/100 g dw. The regression equation was y = 3.826x + 0.0088 ($R^2$ = 0.9988) [35,36].

### 2.8. Statistical Analysis

All tests were performed in triplicate, and the values shown are the means and standard deviations (SD). The univariate statistic test, one-way ANOVA ($p$ = 0.05), combined with Tukey's multiple comparisons tests, was used to assess the physico-chemical characteristics, the bioactive compound content, and the antioxidant capacity of blackberry cultivars. Univariate statistical analysis was performed with Stata 17SE statistical software (StataCorp LLC, 4905 Lakeway Drive, College Station, TX, USA). Principal component analysis (PCA), linear discriminant analysis (LDA), multivariate analysis of variance (MANOVA) ($p$ = 0.05), and hierarchical cluster analysis (HCA) were the multivariate statistical methods that

were used to classify the analyzed blackberry cultivars in accordance with the parameters examined. Multivariate statistical analysis was performed with MATLAB v2022b CWL (MathWorks, 1 Apple Hill Drive, Natick, MA, USA).

## 3. Results and Discussion

### 3.1. Physico-Chemical Parameters of Blackberries

Blackberries are perishable fruits due to their external structure and the very high percentage of moisture in the pulp [37,38].

Table 2 shows the average fresh fruit weight (GrP), height and diameter, and average fruit weight after dehydration (GrD) of the eight different blackberry cultivars as growth and development parameters. Additionally, the moisture content in each blackberry variety was determined (Table 2) by measuring the fruits after they were dried completely out in a 40 °C oven until they stabilized in weight.

**Table 2.** Moisture content (%), fresh fruit weight (GrP), height, diameter, and dried fruit weight (GrD) for the eight blackberry cultivars (results are presented as means ± SD).

| Samples / Parameters | CHES | T_C | EVRG | NAV | L_NESS | THRNFR | OCHIT | WILD |
|---|---|---|---|---|---|---|---|---|
| Moisture content (%) | 68.8 ± 0.6 [a] | 75.6 ± 0.02 [b] | 73.1 ± 0.05 [c] | 71.8 ± 0.08 [d] | 68.3 ± 0.02 [a, c] | 82.7 ± 0.02 [f] | 64.1 ± 0.1 [g] | 79.9 ± 0.03 [h] |
| GrP (g) | 5.89 ± 0.14 [a] | 6.67 ± 0.52 [b] | 5.87 ± 0.14 [a] | 7.14 ± 0.12 [b] | 4.55 ± 0.08 [c] | 9.11 ± 1.16 [d] | 5.55 ± 0.18 [a] | 1.72 ± 0.001 [e] |
| Height (cm) | 2.33 ± 0.35 [a] | 2.4 ± 0.26 [a] | 2.47 ± 0.11 [a] | 2.43 ± 0.49 [a] | 2.33 ± 0.38 [a] | 2.77 ± 0.15 [a] | 2.30 ± 0.26 [a] | 1.03 ± 0.47 [b] |
| Diameter (cm) | 6.67 ± 0.25 [a, d, e] | 6.97 ± 0.23 [a, d, e] | 6.93 ± 0.11 [a, d, e] | 6.77 ± 0.32 [a, d, e] | 5.67 ± 0.51 [b] | 7.20 ± 0.26 [d] | 6.17 ± 0.30 [e] | 3.60 ± 0.37 [c] |

Results are represented as means ± SD; different letters indicate significant differences within the same line ($p < 0.05$); GrP—average weight of a fresh fruit, expressed in g.

Based on the results in Table 2, the percentage of moisture differs according to the variety of fruit studied. The highest moisture percentage was recorded by THRNFR and the lowest by OCHIT. All blackberry varieties showed high moisture percentages; however, the percentage of moisture was lower than in other research in the literature [37,38]. According to the US Department of Agriculture, in addition to other nutrients present in blackberries, a high-water content of approximately 88.2% has been reported [37]. In another study, the moisture content of blackberries was 87.92 ± 0.59 [38].

Agricultural techniques, climate (high temperatures), fruit development stage, size, surface-to-volume ratio, or external structure are only a few of the many factors that could influence fruit moisture. Even amongst cultivars of the same species, all the mentioned factors might result in significant humidity differences [39,40].

From the fresh juice obtained for each variety, the physico-chemical properties of blackberries, including pH, titratable acidity (%), and the amount of total soluble solids (TSS) expressed in °Brix, were determined and are shown in Table 3. Blackberries obtained from the spontaneous flora (WILD) and those grown in cultivation were both assessed for firmness immediately after harvest.

**Table 3.** The physico-chemical parameters (pH, acidity, firmness, and TSS content of eight different blackberry varieties).

| Samples / Parameters | CHES | T_C | EVRG | NAV | L_NESS | THRNFR | OCHIT | WILD |
|---|---|---|---|---|---|---|---|---|
| pH | 3.27 ± 0.014 [f] | 3.65 ± 0.012 [d] | 2.85 ± 0.17 [g] | 3.01 ± 0.012 [e] | 3.84 ± 0.012 [b,c] | 3.98 ± 0.008 [a, b, c] | 4.066 ± 0.018 [a] | 3.95 ± 0.014 [a, c] |
| Acidity (% malic acid) | 6.21 ± 0.60 [b] | 4.37 ± 0.30 [d] | 7.86 ± 0.30 [a] | 7.46 ± 0.6 [a] | 3.18 ± 0.44 [e] | 2.55 ± 0.44 [e] | 2.53 ± 0.054 [e, f] | 3.83 ± 0.12 [d, e] |
| Firmness | 5.28 ± 1.06 [a] | 5.33 ± 1.49 [a] | 4.94 ± 0.58 [a] | 8.56 ± 2.36 [b] | 7.17 ± 1.85 [a] | 5.23 ± 0.91 [a] | 6.06 ± 1.22 [a] | 5.78 ± 1.88 [a] |
| TSS | 14.13 ± 0.05 [d] | 14.33 ± 0.1 [c, d] | 14.4 ± 0.08 [c] | 13.85 ± 0.05 [e] | 11.18 ± 0.09 [f] | 12.2 ± 0.08 [f] | 14.7 ± 0.18 [b] | 18.18 ± 0.1 [a] |

Results are represented as means ± SD; different letters indicate significant differences within the same line ($p < 0.05$); TSS—total soluble solids.

The cultivars EVRG and NAV had the lowest pH values, which were $2.85 \pm 0.17$ and $3.01 \pm 0.01$, respectively, whereas THRNFR, WILD, and L_NESS had the highest pH values. The titratable acidity of blackberry fruits has been recorded to range from $2.53 \pm 0.44\%$ to $7.86 \pm 0.3\%$, depending on the cultivar. No significant differences were found in terms of firmness, with the exception of the NAV cultivar. Regarding the TSS content, statistically significant differences between the wild variety (WILD) and the cultivated ones were found (Table 3).

De Souza, et al., 2014, reported Brazilian blackberries' pH values of $2.99 \pm 0.04$, acidity of $1.51 \pm 0.04$ % and TSS of $10.17 \pm 0.29$ °Brix [38]. Petkovsek, et al., 2021, reported pH values between 2.7 and 3.2 for ´Loch Ness´, ´Navaho´ and ´Thornfree´ blackberries (Slovenia), and TSS recorded values between 10.58 and 14.4 °Brix, for ´Loch Ness´, between 10.74 and 13.30 °Brix for ´Navaho´ and between 7.68 and 9.72 °Brix. Furthermore, the reported weight for the "Loch Ness" variety ranged from 4.85 to 6.9 g, while the weights of the "Navaho" and "Thornfree" varieties were respectively 4.84 and 7.45 g, and 5.26 and 7.07 g [14]. In a recent study, Garazhian, et al., 2020, investigated changes and interactions for fruit weight, total solid solubility, and titratable acidity in four species of Iranian-origin blackberries from the Rubus genus over the course of two years. Analysis has indicated that there was no significant difference between the data collected over a two-year period and the berries' weight, TSS, and titratable acidity values, which ranged from 0.14 to 1.30 g, 7.9 to 17.8 °Brix and 0.36 to 0.83%, respectively [41].

*3.2. The Content of Total Phenols (TPh), Total Flavonoids (TFlav) in Blackberry Fruits and Monomeric Anthocyanins (MAP)*

Blackberries are highly rich in phenolic acids, anthocyanins, and flavonoids [4,38,42], in addition to the typical components such as vitamins and minerals [43].

Figure 2a–c shows the results from the content of TPh, MAP, and TFlav of the blackberry taken in this study.

L_NESS and THRNFR fruits had the highest TPh content among the eight varieties tested (Figure 2a). They exhibited TPh contents of $1830.98 \pm 13.55$ and $1687.14 \pm 62.41$ mg GAE/100 g dw, respectively. The result with the lowest value was EYE $847.25 \pm 54.26$ mg GAE/100 g dw. According to Figure 2c, which compares the TFlav content of samples from eight different varieties of dehydrated blackberries, THRNFR had the highest content ($1545.73 \pm 77.15$ mg QE/100 g dw), followed by sample L_NESS ($1271.38 \pm 25.03$ mg QE/100 g dw), and CHES ($1094.4 \pm 62.98$ mg QE/100 g dw. The OCHIT sample, with a reported value of $638.65 \pm 8.84$ mg QE/100 g dw, was found to have the lowest TFlav level. THRNFR recorded the highest MAP content ($670.03 \pm 29.98$ mg cyanidin-3-glucoside (C3glu)/100 g dw.) among blackberries (Figure 2b). The lowest MAP content was recorded in NAV, with a reported value of $61.86 \pm 1.32$ mg C3glu/100 g dw. There were no statistically significant MAP differences ($p < 0.05$) between the WILD, EVRG, and NAV varieties, respectively (Figure 2b).

The TPh, TFlav, and MAP values recorded for dried blackberries were found to be higher than those reported for fresh fruit when compared to research in the literature. For instance, de Souza, et al., (2014) reported that the TPh content in blackberries was $850.52 \pm 4.77$ mg GAE/100 g f.w, the TFlav was $87.03 \pm 4.85$ mg CE/100 g f.w, and the MAP content was 58, $61 \pm 2.19$ mg C3glu/100 g of f.w [38].

The anthocyanin and total phenolic content of thornless blackberry varieties such as "Chester" and "Triple Crown" were reported by Wang and Lin in 2000. In comparison to unripe fruits (3.9 1.1 mg C3glu/100 g dw and 7.1 1.7 mg C3glu/100 g dw, respectively), the number of anthocyanins reported in the case of "Chester" was 912.5 27.3 mg C3glu/100 g dw whereas that of "Triple Crown" was 794.6 12.5 mg C3glu/100 g dw [42].

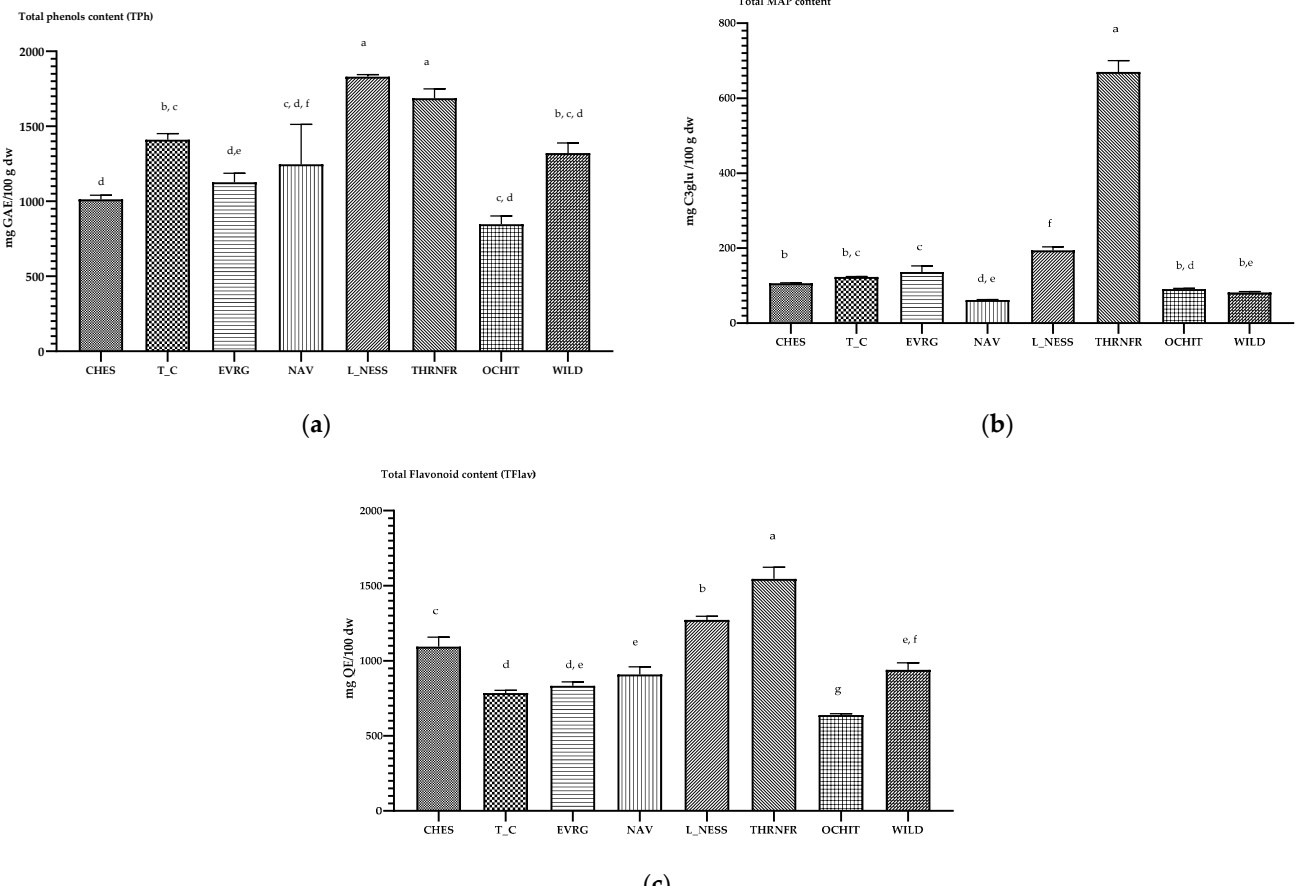

**Figure 2.** The bioactive compounds of blackberry varieties. (**a**) Total phenols content (TPh) expressed as milligram gallic acid equivalent (GAE)/100 g dry weight (dw); (**b**) total monomeric anthocyanin content (MAP) expressed as cyanidin-3-glucoside (mg C3glu/100 g dw.); and (**c**) total flavonoid content (TFlav) expressed as milligrams of quercetin equivalent (QE)/100 g dw. All results represent the mean $\pm$ SD of triplicate analysis. Different letters indicate a significant difference ($p < 0.05$) between varieties.

Also, Kolniac-Ostek, et al., 2015, studied five thornless blackberry cultivars and five thorny blackberry cultivars, reporting that thornless blackberry fruits had higher anthocyanin content (mean = 171.23 mg/100 g fw.) compared to spiny varieties [44].

In another study, Sariburun, et al., 2010, reported the number of polyphenols (2786.8 $\pm$ 21.9 mg GAE/100 g fw.) and flavonoids (82.2 $\pm$ 1.3 mg CTE/100 g fw.) from the methanolic extract from fresh fruits of *R. fruticosus* [45]. Our findings are consistent with those in the literature [20] regarding the total amount of phenolics and anthocyanins in *R. laciniatus* ('Evergreen').

There are currently limited investigations of the phytochemical profile and information on the antioxidant capacity of blackberry fruits grown from the *R. fruticosus* and *R. laciniatus* species, with significant differences in their TPh, TFlav, or MAP contents [14,17,20,46–49]. Differences in the content of bioactive compounds present in fruits obtained for the same cultivar or for varieties belonging to the same genus can arise from various factors, such as cultivation techniques, the difference in climatic conditions (temperature, solar radiation, precipitation), soil composition, and the different methods of extraction of bioactive compounds [50–52].

### 3.3. Anthocyanins Identified from Blackberry Fruits by HPLC-PDA

Anthocyanin pigments are responsible for the color of the fruit; specialized studies have also highlighted the role of these compounds in the prevention and treatment of

numerous ailments [37,48,53]. Therefore, the anthocyanins from the eight varieties of dehydrated blackberry fruits were isolated by SPE and analyzed by HPLC-PDA; the quantitative results are presented in Table 4.

**Table 4.** Anthocyanins from eight blackberry cultivars separated and quantified (mg cyanidin equivalent/g dw) by HPLC-PDA.

| SAMPLES | CHES | T_C | EVRG | NAV | L_NESS | THRNFR | OCHIT | WILD |
|---|---|---|---|---|---|---|---|---|
| Cyanidin-3-glucoside | 7.51 ± 0.17 [a] | 58.73 ± 4.34 [b, c] | 54.48 ± 4.01 [c, f] | 13.54 ± 0.88 [a, f] | 87.76 ± 5.74 [d] | 329.26 ± 7.64 [e] | 21.48 ± 1.71 [a] | 43.99 ± 3.15 [f] |
| Cyanidin-3-O-arabinoside | 1.45 ± 0.1 [a, f] | 1.02 ± 0.1 [a] | 5.69 ± 0.19 [b, e] | 2.34 ± 0.02 [f] | 8.04 ± 0.60 [c] | 17.40 ± 0.09 [d] | 1.54 ± 0.01 [a, f] | 5.87 ± 0.39 [e] |
| Cyanidin-3-O-(malonyl)glucoside | 0.64 ± 0.04 [f] | 1.21 ± 0.01 [e] | 2.04 ± 0.02 [c] | 1.69 ± 0.01 [d] | 3.46 ± 0.02 [b] | 12.83 ± 0.09 [a] | nd | nd |
| Cyanidin-3-O-(dioxalyl)glucoside | 0.75 ± 0.001 [g] | 1.81 ± 0.01 [e] | 3.39 ± 0.02 [c] | nd | 5.58 ± 0.03 [b] | 17.98 ± 0.13 [a] | 1.05 ± 0.01 [f] | 2.37 ± 0.04 [d] |
| Cyanidin-3-rutinoside | 0.93 ± 0.0001 [a, g] | 2.89 ± 0.02 [b] | 4.34 ± 0.02 [c] | 3.33 ± 0.02 [d] | 17.76 ± 0.13 [e] | 14.92 ± 0.11 [f] | 0.84 ± 0.002 [g] | 2.13 ± 0.01 [h] |
| TOTAL | 11.28 ± 0.07 [a] | 65.67 ± 1.93 [b, c, f] | 69.95 ± 1.80 [c, f] | 20.84 ± 0.39 [a] | 122.60 ± 2.49 [d] | 392.42 ± 3.37 [e] | 24.92 ± 0.76 [a] | 54.36 ± 1.37 [f] |

Values are presented as mean ± SD. Different lowercase letters indicate a significant difference ($p < 0.05$) between harvest stages. Rt—Retention time; dw—dry weight; nd—not detected.

Cyanidin-3-glucoside was the main anthocyanin found in all blackberry cultivars (Table 4); THRNFR cultivar detected the greatest quantity, 86.64%, more than in the wild blackberry (WILD), and CHES found the lowest amount, 7.51 ± 0.21 mg cyanidin equivalent/g dw. These findings are consistent with previous research that indicate cyanidin 3-glucoside is the main anthocyanin in blueberry fruits [13,15]. The cultivar L_NESS reported the highest level of cyanidin-3-rutinoside, contributing to 14.51% of all the anthocyanins found in this cultivar and being 88.01% greater than the WILD, while cultivar OCHIT recorded the lowest concentration. Anthocyanins such as cyanidin-3-O-(dioxalyl) glucoside were not found in the cultivar NAV, while cyanidin-3-O-(malonyl) glucoside was not found in the cultivar OCHIT or wild blackberries (WILD). All known blackberry varieties included cyanidin-3-O-arabinoside, although T_C had the lowest concentration (82.62% less than the wild variety).

Blackberries contain anthocyanins in glycosylated form, which are responsible for the purple, blue, or red colors [15,37]. Approximately 94% of the anthocyanins present in blackberries are presented in non-acylated form, of which approximately 90% are monoglycosides, while approximately 10% are found as diglycosides [37,54,55]. Different studies in the specialized literature have demonstrated that the anthocyanins present in blackberry fruits are derived from cyanidin, and anthocyanins such as cyanidin 3-glucoside, cyanidin-3-O-arabinoside, cyanidin-3-O-(malonyl)glucoside, cyanidin-3-O-(malonyl)glucoside, and cyanidin-3-rutinoside, have been identified [56–59]. Blackberry cyanidin derivatives may vary according to the variety, environment, cultivation region, and fruit age, but primarily due to genetic variations [37,57,60]. According to research by Cho, et al., (2004), the distribution of cyanidin aglycones differed with variety, ranging from 75% to 84% for cyanidin 3-glucoside, 1% to 12% for cyanidin 3-rutinoside, 4% to 8% for cyanidin-3-O-(dioxalyl)glucoside, 3% to 8%, and 2% to 3% for cyanidin-3-O-(malonyl)glucoside [59].

Many studies have shown that blackberry anthocyanins are powerful antioxidants [4,13,61]. The color of anthocyanins depends on their structure, the acidity of the environment, and the presence of copigments [37,62]. Blackberries produce more anthocyanin pigments when they ripen, and the color of these pigments varies depending on the class to which they belong [15,48,63–65]. Many variables—including the variety, the agronomic techniques used in cultivation, the maturity stage of collection, as well as the geological and climatic conditions of the area from which the fruits are harvested—affect the color or

the natural pigments present in blackberries; therefore, these parameters are important to study [15,63–65].

### 3.4. Antioxidant Capacity of Blackberry Fruits

Blackberries are a good source of natural antioxidants. The total amount of phenols, anthocyanins, and flavonoids in blackberries influences their antioxidant capacity [4,38]. Numerous methods have been developed for in vitro measurement of the antioxidant capacity of natural products [16,46]. Three distinct assays were used in this work to measure the antioxidant capacity of extracts from seven cultivated blackberry cultivars and one from the wild flora. The results are presented in Figure 3a–c.

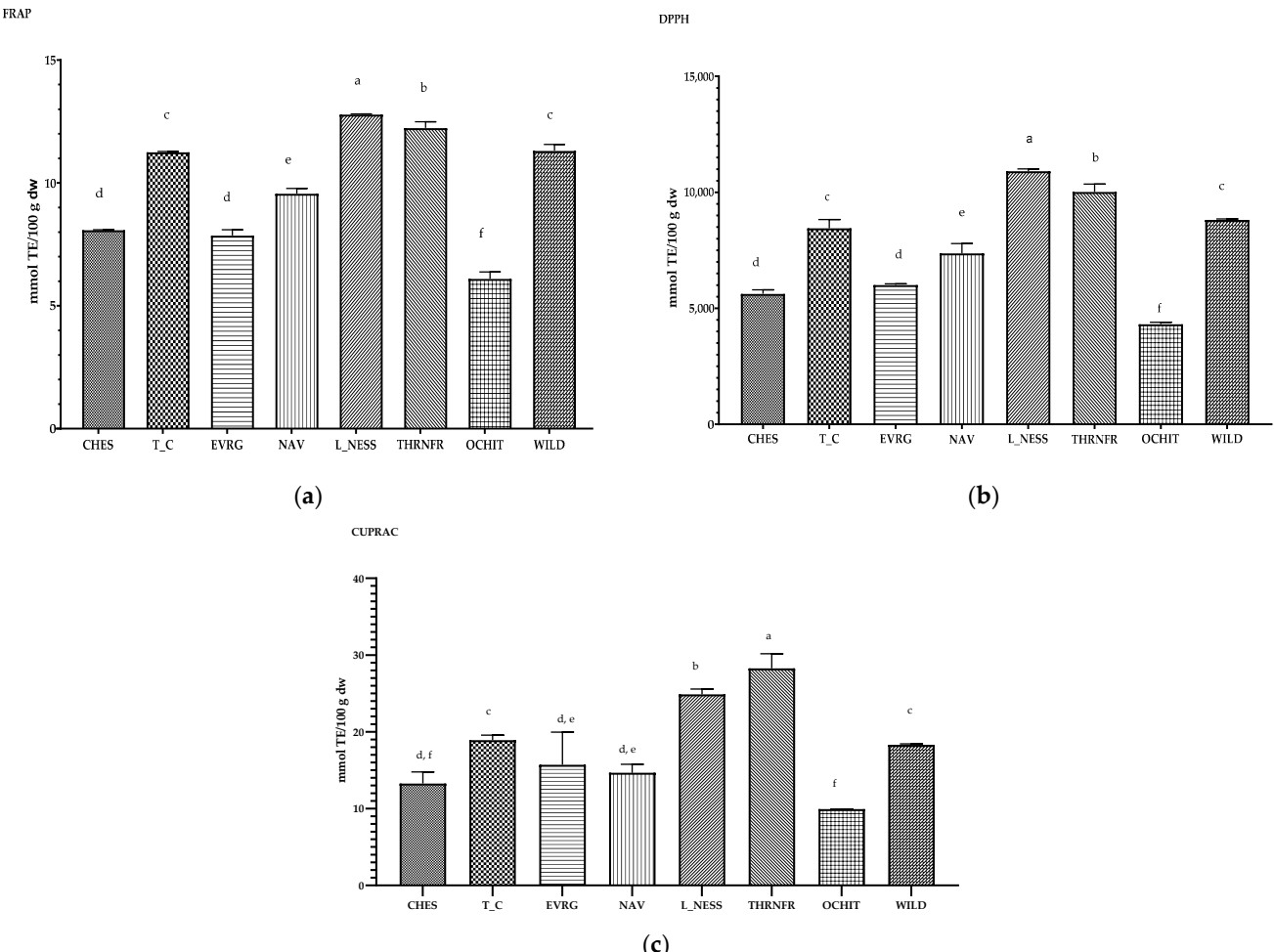

**Figure 3.** The antioxidant capacity of eight blackberry varieties determined by three different methods; (**a**) FRAP (ferric reducing ability of plasma) expressed in mmol Trolox Equivalents (TE)/100 g dw; (**b**) DPPH (2,2, diphenyl-picryl-hydrazyl) expressed in mmol TE/100 g dw; and (**c**) CUPRAC (Cupric Ion Reducing Antioxidant Capacity) expressed in mmol TE/100 g dw. All results represent the mean ± SD of triplicate analysis. Different letters indicate a significant difference ($p < 0.05$) for each variety.

Methods based on single electron transfer (SET), including FRAP, DPPH, and CUPRAC, are used to measure antioxidant activity [24,66,67].

Significant differences were obtained between the eight blackberry varieties in terms of antioxidant capacity. The highest value obtained in the case of the FRAP method, and the DPPH test, was in the case of L_NESS blackberries (12.79 ± 0.02 and, respectively, 10,962.65 ± 88.96 mmol TE/100 g dw), followed by THRNFR and WILD. The antioxidant

capacity of blackberries from the spontaneous flora (WILD) determined, both by the FRAP test (11.32 ± 0.26 mmol TE/100 g dw) and DPPH (8818.5 ± 35.6 mmol TE/100 g dw), recorded significantly higher values, compared to other varieties presented in Figure 3a,b. On the other hand, the CUPRAC method's results showed that the THRNFR blackberry cultivar had the maximum antioxidant capacity (28.27 ± 1.89 mmol TE/100 g dw), followed by L_NESS (24.88 ± 0.69 mmol TE/100 g dw) (Figure 3c). The OCHIT blackberry cultivar has the lowest antioxidant capacity, as measured by FRAP (6.1 ± 0.28 mmol TE/100 g dw), DPPH (4321.11 ± 80.06 mmol TE/100 g dw), and CUPRAC (9.93 ± 0.03 mmol TE/100 g dw.).

The FRAP method was initially used to evaluate the antioxidant capacity of plasma, and later, it was tested on fruits and successfully used to evaluate the antioxidant-reducing power of different extracts [33]. Siriwoharn and Wrolstad 2004, tested the antioxidant capacity by the FRAP method, both from the whole fruits of *R. laciniatus*, variety ´Evergreen´, as well as from seeds and seedless fruits. The highest antioxidant capacity was reported in the seeds (197 ± 163 µmol TE/g), about three times higher than in the whole fruits of the cultivar ´Evergreen´ [20]. The antioxidant capacity of blackberry fruits varies greatly depending on the area, climatic conditions, variety, fruit maturity stage, extraction solvents, and methods used [4,20,42].

In recent years, various studies have demonstrated that the high level of natural antioxidants in blackberry fruits have multiple benefits for human health, thus considerably increasing the interest in the cultivation and development of as many varieties as possible [3,11,48,68–70].

Heinonen, et al., 1998, demonstrated, in vitro, that berries of various varieties, including blackberry, have remarkably high scavenging activity against chemically generated superoxide radicals [71]. Additionally, the antioxidant capacity measured for the fresh fruits of *R. fruticosus* by the DPPH method recorded values between 7.19 ± 0.06 µmol TE/g fw. [47] and 177.11 ± 3.17 µmol TE/g fw. [45]. Another study by Suriburum, et al., (2010) investigated the antioxidant capacity of blackberries using the CUPRAC method, and the results showed a value of 127.15 ± 0.57 mol TE/g fw. [45].

The seven blackberry cultivars that were harvested from the same region were characterized by a varied number of bioactive compounds from the polyphenol class and, in particular, by a different profile and content of anthocyanins. There are several factors that could be responsible for these variations, but genetic factors are highly relevant for the content of these secondary metabolites [72].

### 3.5. Multivariate Analysis

Table 5 shows the correlation matrix between physico-chemical and biochemical parameters, including the content of total phenols, flavonoids, and monomeric anthocyanins. Physical parameters correlation exhibits very low values (i.e., weak correlation) and only a limited value of statistically significant correlations ($p$ = 0.05). A strong correlation is encountered between pH and Acidity parameters (an expected result) and is statistically significant. From these results, pH with Acidity presents statistically significant strong and negative correlations; also, MAP with TPh, TFLAV and all antioxidant capacities present statistically significant average and positive correlations. Furthermore, between MAP and C3glu, C3Oara, C3Omal, C3Odio, and C3Orut, there are statistically significant strong and positive correlations. However, statistically significant strong correlations are performed between all biochemical parameters.

**Table 5.** Correlation matrix for physico-chemical and biochemical parameters, as Pearson coefficient. Boldface numbers denote statistically significant correlations ($p = 0.05$).

| R | pH | Acid | Firm | TPh | TFlav | TSS | MAP | FRAP | DPPH | CUPRAC | Moisture | C3glu | C3Oara | C3Omal | C3Odio | C3Orut |
|---|---|---|---|---|---|---|---|---|---|---|---|---|---|---|---|---|
| pH | **1.000** | **−0.961** | −0.124 | 0.290 | 0.228 | 0.022 | **0.357** | 0.341 | 0.349 | **0.446** | −0.176 | **0.394** | 0.342 | 0.254 | **0.389** | 0.313 |
| Acid | **−0.961** | **1.000** | 0.107 | −0.347 | −0.288 | 0.124 | **−0.441** | **−0.351** | **−0.374** | **−0.499** | 0.086 | **−0.469** | **−0.403** | −0.344 | **−0.467** | **−0.413** |
| Firm | −0.124 | 0.107 | **1.000** | 0.128 | 0.092 | −0.178 | −0.154 | 0.213 | 0.210 | 0.059 | 0.009 | −0.145 | −0.072 | −0.068 | −0.150 | 0.168 |
| TPh | 0.290 | −0.347 | 0.128 | **1.000** | **0.696** | **−0.505** | **0.549** | **0.916** | **0.952** | **0.920** | 0.323 | **0.603** | **0.648** | **0.596** | **0.605** | **0.849** |
| TFlav | 0.228 | −0.288 | 0.092 | **0.696** | **1.000** | **−0.560** | **0.804** | **0.662** | **0.691** | **0.818** | **0.704** | **0.780** | **0.833** | **0.823** | **0.813** | **0.798** |
| TSS | 0.022 | 0.124 | −0.178 | **−0.505** | **−0.560** | **1.000** | **−0.494** | −0.281 | **−0.356** | **−0.472** | **−0.526** | **−0.447** | **−0.396** | **−0.567** | **−0.469** | **−0.749** |
| MAP | **0.357** | **−0.441** | −0.154 | **0.549** | **0.804** | **−0.494** | **1.000** | **0.468** | **0.497** | **0.746** | **0.783** | **0.988** | **0.919** | **0.983** | **0.987** | **0.680** |
| FRAP | 0.341 | **−0.351** | 0.213 | **0.916** | **0.662** | −0.281 | **0.468** | **1.000** | **0.984** | **0.913** | 0.173 | **0.541** | **0.573** | **0.503** | **0.525** | **0.709** |
| DPPH | 0.349 | **−0.374** | 0.210 | **0.952** | **0.691** | **−0.356** | **0.497** | **0.984** | **1.000** | **0.934** | 0.235 | **0.568** | **0.629** | **0.539** | **0.563** | **0.793** |
| CUPRAC | **0.446** | **−0.499** | 0.059 | **0.920** | **0.818** | **−0.472** | **0.746** | **0.913** | **0.934** | **1.000** | **0.431** | **0.797** | **0.806** | **0.765** | **0.795** | **0.862** |
| Moisture | −0.176 | 0.086 | 0.009 | 0.323 | **0.704** | **−0.526** | **0.783** | 0.173 | 0.235 | **0.431** | **1.000** | **0.743** | **0.785** | **0.845** | **0.770** | **0.553** |
| C3glu | **0.394** | **−0.469** | −0.145 | **0.603** | **0.780** | **−0.447** | **0.988** | **0.541** | **0.568** | **0.797** | **0.743** | **1.000** | **0.939** | **0.977** | **0.993** | **0.700** |
| C3Oara | 0.342 | **−0.403** | −0.072 | **0.648** | **0.833** | **−0.396** | **0.919** | **0.573** | **0.629** | **0.806** | **0.785** | **0.939** | **1.000** | **0.923** | **0.963** | **0.780** |
| C3Omal | 0.254 | −0.344 | −0.068 | **0.596** | **0.823** | **−0.567** | **0.983** | **0.503** | **0.539** | **0.765** | **0.845** | **0.977** | **0.923** | **1.000** | **0.975** | **0.729** |
| C3Odio | **0.389** | **−0.467** | −0.150 | **0.605** | **0.813** | **−0.469** | **0.987** | **0.525** | **0.563** | **0.795** | **0.770** | **0.993** | **0.963** | **0.975** | **1.000** | **0.738** |
| C3Orut | 0.313 | **−0.413** | 0.168 | **0.849** | **0.798** | **−0.749** | **0.680** | **0.709** | **0.793** | **0.862** | **0.553** | **0.700** | **0.780** | **0.729** | **0.738** | **1.000** |

C3glu—Cyanidin-3-glucoside; C3Oara—Cyanidin-3-O-arabinoside; C3Omal—Cyanidin-3-O-(malonyl)glucoside; C3Odio—Cyanidin-3-O-(dioxalyl)glucoside; C3Orut—Cyanidin-3-rutinoside.

In this study, sixteen parameters were measured, which can be gathered for each sample as a multivariate profile. The profiles were compared with a multivariate statistical sequence, with the purpose of extracting the multivariate correlations between the parameters and the samples and the grouping of the samples (i.e., the clustering). The multivariate statistical sequence consisted of principal component analysis (PCA), linear discriminant analysis (LDA), multivariate analysis of variance (MANOVA) ($p = 0.05$), and hierarchical cluster analysis (HCA). Among these statistical methods, only MANOVA can perform results with statistical significance ($p = 0.05$); in consequence, all the multivariate results and the corresponding conclusions will have 95% accuracy.

Results of the PCA are shown in Table S1 and Figure 4. The total variance of principal components with an eigenvalue higher than unity are PC1–PC3, with a value of 87.27%. For biological systems, this value is satisfactory for drawing the statically proper conclusions. However, the analysis that follows will consider all the principal components for the calculation of the LDA and MANOVA results. Principal component results represented in the PC1–PC3 principal axis frame as 2D and 3D biplots contain both spatial distribution of the variables and samples (Figure 4). The variables distribution is represented with vectors starting from the principal axes' frame center point. The vectors end points out the direction with the highest content of the corresponding variable. The opposite direction points out the direction with the lowest content of the corresponding variable. In consequence, the samples that present their geometric projection on one vector end will perform high content of the corresponding variable, and vice versa.

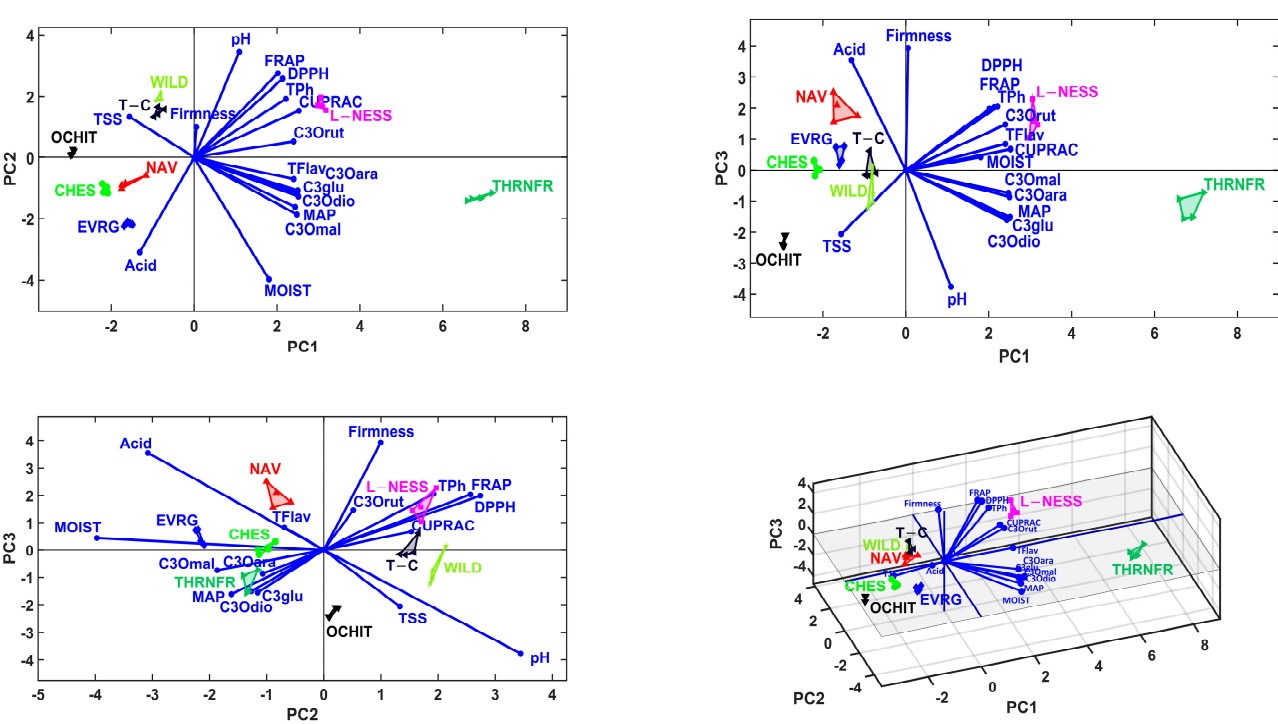

**Figure 4.** Principal component analysis (PCA) results represented as 2D and 3D biplot graphs.

Geometrical variable vector arrangement can prescribe the multivariate correlations between the variables and, furthermore, the variable grouping. Biplots from Figure 4 suggest five singleton groups: Firmness, Moisture, Acidity, TSS pH, and C3Orut, and two multiple variable groups. The first multiple variables group consists of FRAP, DPPH, TPh, and CUPRAC; the second group consists of TFlav, MAP, C3glu, C3Oara, C3Omal, and C3Odio. These variable groups determine the samples' spatial distributions. Cultivar samples THRNFR and L_NESS have high levels of antioxidant capacity (FRAP, DPPH, TPh, and CUPRAC) together with TFlav, TPh, MAP, C3glu, C3Oara, C3Omal, and C3Odio, also for Moisture, pH, and medium level for Firmness. Cultivars OCHIT, WILD, and T_C

perform high levels of TSS and pH, but with the lowest levels of the other variables. Cultivar samples EVRG, CHES, and NAV perform high levels for variables Acidity, Firmness, and Moisture, but with the lowest levels of the other variables (see Figure 4). In order to determine that this kind of samples grouping (i.e., based on PCA results) generates the same clusters, the LDA, MANOVA ($p = 0.05$) and HCA statistical multivariate methods were applied to the principal coordinates of the samples (i.e., the PC of the samples consisted as input data for mentioned methods).

The results of the effective clustering method are shown in Table S2 and Figure 5. The LDA method generates the canonical coordinates that generally maximize the Euclidean distances between the samples (Figure 5). These canonical coordinates are used by MANOVA ($p = 0.05$) to calculate the statistical significance values from sample profiles pairwise comparisons (Table S2). As it can be noticed, all these values are higher than the $p = 0.05$, a fact that validates each sample to be a singleton cluster.

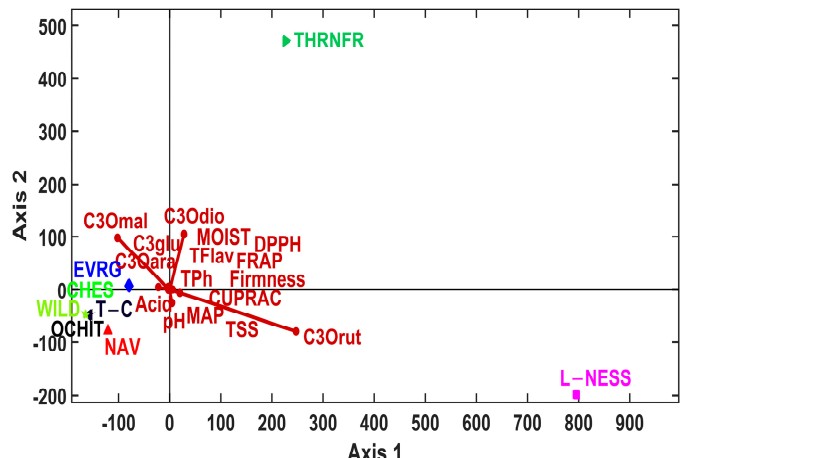
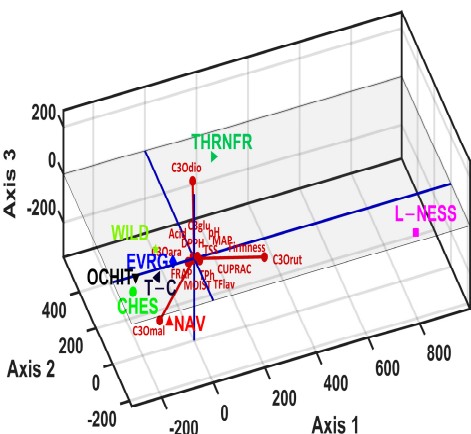

**Figure 5.** Linear discriminant analysis (LDA) results represented as 2D and 3D biplot graphs.

The heatmap emphasizes that by choosing only the first three canonical axes (e.g., the first three principal axes), the clustering conclusions are still valid (Figure 6, from Axis 4, the black color means null contributions to the clustering process).

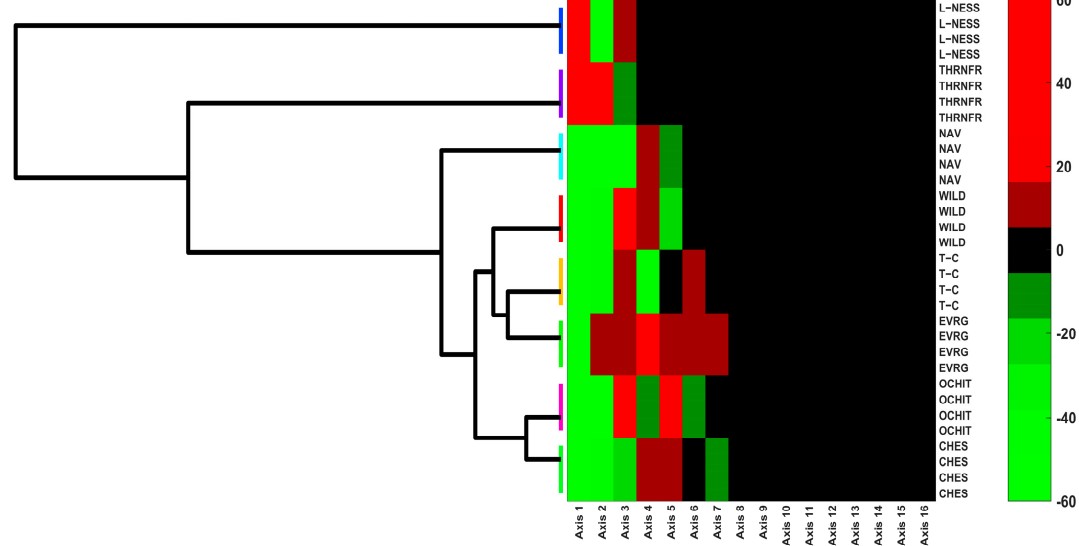

**Figure 6.** Graphical clustering information of analyzed samples organized as a heatmap.

The graphical LDA biplots with clustering information are displayed in Figure 7.

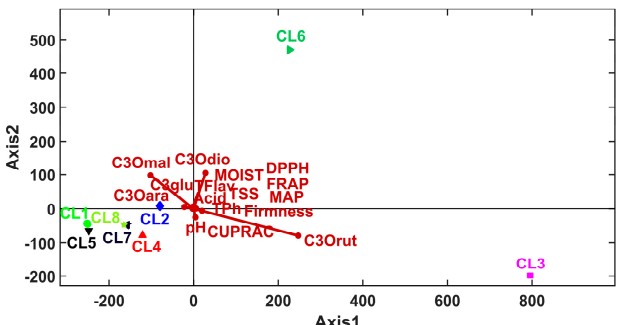
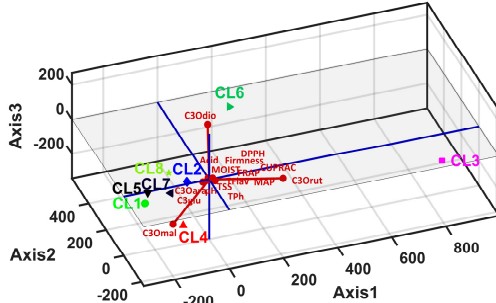

**Figure 7.** Linear discriminant analysis (LDA) results represented as 2D and 3D biplot graphs with MANOVA and HCA-validated clusters. The 8 clusters, validated by the MANOV (*p* = 0.05) and HCA multivariate methods, were generated: Cluster 1 (NAV), Cluster 2 (EVRG), Cluster 3 (CHES), Cluster 4 (OCHIT), Cluster 5 (WILD), Cluster 6 (T_C), Cluster 7 (L_NESS), Cluster 8 (THRNFR).

## 4. Conclusions

In this study, blackberries from the spontaneous flora and seven blackberry cultivars grown on a farm in NW Romania were compared in terms of their physico-chemical properties, phytochemical composition, and antioxidant capacity. The biggest blackberry cultivars in terms of fruit size are THRNFR, followed by NAV and T_C, and wild blackberries are the smallest. The pH of the mulberry cultivars ranged from 2.85 to 4.066, and with the exception of the NAV cultivar, there are no appreciable differences amongst the cultivars in terms of firmness. The results showed that, of the seven cultivars, cultivars THRNFR and L_NESS significantly differed from the others with respect to the number of bioactive compounds and the antioxidant capacity.

In addition, the multivariate analysis emphasizes the variables grouping: one group of antioxidant capacities (FRAP, DPPH, TPh, and CUPRAC) and MAP, one group of Tflav, MAP, C3glu, C3Oara, C3Omal, and C3Odio and five singleton groups: Firmness, Moisture, Acidity, TSS, and pH, and two multiple variable groups. Cultivar samples THRNFR and L_NESS have the multivariate profile with the highest levels of the antioxidant capacities, TFLAV, MAP, C3glu, C3Oara, C3Omal, C3Odio, firmness, and pH, and thus they can be denoted as functional products. At the opposite direction in the principal components PC1 and PC2 plane, the cultivar samples OCHIT, T_C and WILD perform highest levels for TSS and pH, meaning that they are the "sweetest" blackberry samples.

In this way, the multivariate analysis performed a scientific and commercial classification of the analyzed blackberry cultivars. The sweet and high firmness level samples should be used in the bakery industry segment. The sweet and soft samples should be used in the beverages and jam industry. A significant finding is that the THRNFR and L_NESS cultivars can be considered functional food that can be used as food supplements or in the pharmaceutical sector.

As future perspectives, in-depth studies on the phytochemical composition of the new blackberry cultivars and varieties introduced in Romania are needed to identify the bioactive compounds responsible for the antioxidant capacity.

**Supplementary Materials:** The following supporting information can be downloaded at: https://www. mdpi.com/article/10.3390/horticulturae9050556/s1, Table S1: Summarized results from PCA method; Table S2: Statistical significance values from sample profiles pairwise comparisons calculated with MANOVA method (*p* = 0.05).

**Author Contributions:** Conceptualization, S.I.V. and A.R.M.; Validation, A.R.M., R.B. and S.I.V.; Formal Analysis, A.C.T.; Investigation, A.R.M., S.I.V. and I.S.; Resources, S.I.V. and R.B.; Writing—Original Draft Preparation, A.R.M.; Writing—Review & Editing, S.I.V. and A.R.M.; Visualization, A.R.M. and M.B.; Supervision, S.I.V.; Project Administration, S.I.V.; Funding Acquisition, R.B. and I.S. All authors have read and agreed to the published version of the manuscript.

**Funding:** APC was funded by the University of Oradea.

**Data Availability Statement:** Not applicable.

**Acknowledgments:** The authors acknowledge the support provided by University of Oradea. The grant "Excellence scientific research related to priority fields with valorization through technology transfer: INO-TRANSFER-UO-2nd edition," Project no. 250/08.11.2022, provided the chemicals and material support for this study.

**Conflicts of Interest:** The authors declare no conflict of interest.

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
