# Peer review of "Bioactive Compounds and Antioxidant Capacity of Several Blackberry (Rubus spp.) Fruits Cultivars Grown in Romania"

_horticulturae, doi:10.3390/horticulturae9050556_

Round 1

Reviewer 1 Report

The article "Bioactive compounds and antioxidant capacity of several blackberry (Rubus spp.) fruits cultivars grown in Romania" has been reviewed.

According to what has been reviewed, a large amount of laboratory and statistical work can be observed, which makes this manuscript relevant. Some small errors have been noticed regarding the edition of some figures and wording. I consider that the work is well developed and that the results presented are interesting due to the analyzes carried out on each of the samples.

The comments that are recommended to be taken into account can be found in the attached file.

Reviewer 2 Report

The manuscript entitled “Bioactive compounds and antioxidant capacity of several blackberry (Rubus spp.) fruits cultivars grown in Romania” is a well structured scientific paper, characterized blackberry fruits according to a multimethodological approach. The paper is good and well fit with Horticulturae journal.

Therefore, there are some minor comments I suggest before its publication.

Lines 12-13: There is An overlap of email addresses between the lines of the abstract. Please revise it

Lines 31-32: I would add in the keywords also multivariate analysis used in the paper.

Lines 34-44: I would add some specific data for production and consumption of these fruits, thus focusing on Romania.

Lines 58-68: These section should be implemented also indicating quantitative range and relative percentage of the main bioactive compounds found in these samples. In addition authors should also reference in which part of the fruits anthocyanidins and other polyphenols are specifically originated.

Materials and Methods

I suggest to add a paragraph only for chemicals before plant materials

Paragraph 2.1: Even if the section is well structured I suggest to encompass all information about each sample in a table so that content is clearer and more direct.

Lines 283: please provide the software used for statistical analysis.

Results 

Considering FIG.4, samples L_NESS and THRNFR are the only samples to have a different trend among all three assays performed. What could this be due to? I would suggest adding an explanation.

I think table 6 and table 8 are superfluous since the info presented is already contained in the text

Reviewer 3 Report

The paper presents data about different varieties of blackberries grown in Romania.

In my opinion the authors have mixed two objectives which make the work unclear:

firstly they want to evaluate the differences between different blackberry cultivars on the basis of a series of parameters that they have determined, with an accurate statistical analysis (but why not also include the anthocyanin data obtained in HPLC in the statistical evaluation?, see L 565) ; secondly they insert evaluations, for example on the colour, after the drying treatment of the product. In my opinion this part creates confusion with respect to the first objective and in my opinion it would require to separate this evaluation from the previous one and not present everything in the same work. Or if the authors feel it is necessary to keep all the material together, I suggest treating the dehydrated samples independently of the fresh samples.

English needs a thorough review as some terms are used improperly or the construction of sentences is not simple, I am not qualified for a systematic review, but I have noticed this difficulty in understanding the text.

An element that I think is useful to verify is the way of determining the humidity of the samples. It seems to me that heating to 40°C alone is not sufficient to obtain the removal of all the water, in fact the authors obtain lower water values ​​than these reported in the literature. I think that it would be advisable to obtain the moisture parameter with a higher heating method (105°C) and check if it corresponds, for the same sample, to what was obtained at 40°C. Furthermore, it seems that the dehydration process was obtained on whole fruit, while to determine the humidity, it would be more appropriate to work on a homogenate sample, in order to destroy the structure of the fruit.

Furthermore, in Table 1 the fruit weight data after dehydration process are reported, but these data are not discussed, I would suggest to eliminate this row from the table.

Finally, it seems to me very strange to report the evaluation of the color analysis only on dehydrated fruit and not on fresh fruits

Other correction

L16 correct fructicosus in fruticosus

L75-84: a complete list of molecules identified in previous paper is not necessary

L90: being DPOPH a radical is better to indicate it as DPPH°

L 163: is suppose is blackberries and not mulberries

L173: how was obtained the drop used for TSS evaluation?

L176; how was determined the equivalent point and the quantity of sample is not indicated

L179: the autors used a UV spectrophotometer for the evaluation of color parameters? If is correct, other information must be added: wavelength of different colors lecture etc.

L 206, 239, : the S.I. abbreviation for liter is L and not l

L221: correct was in were

L230: correct eluted in dissolved

L246: correct pick in peak

L263: TE change in Trolox equivalent (is the first time that you use the abbreviation)

L254-274; the regression curves used for obtain the TE values may be added

L276: correct run in performed

L314: correct in psysico-chemical

L335: correct different in difference

L341: is not clear for me the choice criterion adopted.

 Fig 3: insert in the figure also the parameter presented (Total phenolics, MAP, TFlav) in order to clearly understand the figure (such us in Figure 4, in this case please improve the dimensions of the test name)

L464-488: don’t use the s for the plural when you cite the different anthocyanins (rutinoside and not rutinosides, for example)

L518: the values of TE around 10000 mmol/100 g are correct, these data seems over-literature range)

Title Table 5: remove “and the statistical significance p-value”, not present in the table

Table 6: not indispensable, the information is reported in the text (you can add it in supplementary material.

Table 7: same consideration of Table 7, in this case the information is present in fig 7 and 8

I suggest also to remove the Fig 7 and maintain only the Fig 8.

Table 8: also this table is not important, the information can be added in the legend of Figure 9

L638: remove the second “cultivars”

L659-661 remove this statement, no information in your paper is related to this point

L692: Press CU is not an Author!

In references botanical names must be in Italic and the species name without capital letter.

Round 2

Reviewer 3 Report

The authors improved the paper following the suggestions proposed by the referee